 PLOS | ONE

# Spectral measure of color variation of black-orange-black (BOB) pattern in small parasitoid wasps (Hymenoptera: Scelionidae), a statistical approach

Rebeca Mora-Castro[1,2,3]*, Marcela Hernández-Jiménez[2,4], Marcela Alfaro-Córdoba[5,6], Esteban Avendano[2,4], Paul Hanson-Snortum[3]

**1** Centro de Investigación en Biología Celular y Molecular, Universidad de Costa Rica, San José, Costa Rica, **2** Centro de Investigación en Ciencia e Ingeniería de Materiales, Universidad de Costa Rica, San José, Costa Rica, **3** Escuela de Biología, Universidad de Costa Rica, San José, Costa Rica, **4** Escuela de Física, Universidad de Costa Rica, San José, Costa Rica, **5** Centro de Investigación en Matemática Pura y Aplicada, Universidad de Costa Rica, San José, Costa Rica, **6** Escuela de Estadística, Universidad de Costa Rica, San José, Costa Rica

* rebeca.mora@ucr.ac.cr

**Data Availability Statement:** All data and code files are available from the public repo https://github.com/malfaro2/Mora_et_al. Following the

## Abstract

Small parasitoid wasps are abundant and extremely diverse, yet their colors have not been analyzed. One of the more common color patterns observed in these wasps is a black-orange-black pattern, which is especially common among neotropical species of Scelionidae ranging in size from 2 to 10 mm. Due to the methodological challenges involved in extracting and analyzing pigments from small-sized insects, other methods for examining colors need to be explored. In this work, we propose the use of microspectrophotometry in combination with statistical analysis methods in order to 8 study the spectral properties in such cases. We examined 8 scelionid genera and 1 genus from a distantly related family (Evaniidae), all showing the black-orange-black pattern. Functional Data Analysis and statistical analysis of Euclidean distances for color components were applied to study color differences both between and within genera. The Functional Data Analysis proved to be a better method for treating the reflectance data because it gave a better representation of the physical information. Also, the reflectance spectra were separated into spectral color component contributions and each component was labeled according to its own dominant wavelength at the maximum of the spectrum: Red, Green and Blue. When comparing spectral components curves, the spectral blue components of the orange and black colors, independent of the genera being compared, result almost identical, suggesting that there is a common compound for the pigments. The results also suggest that cuticle from different genera, but with the same color might have a similar chemical composition. This is the first time that the black and orange colors in small parasitoid wasps has been analyzed and our results provide a basis for future research on the color patterns of an abundant but neglected group of insects.

suggestion from one reviewer we tried to upload the repository contents in a zip file to the submission system, but it seems that the file size exceeds the limit (zip file is 86 MB). The GitHub repository is updated.

**Funding:** This project was supported by the University of Costa Rica (https://vinv.ucr.ac.cr/) grant No.111-B2-A51 (RM-C) and 801 A5B50 (RM-C). The funders had no role in study design, data collection and analysis, decision to publish, or preparation of the manuscript.

**Competing interests:** The authors have declared that no competing interests exist.

# Introduction

The reflection, absorption or scattering of specific wavelengths of light by surface structures, the deposition of different chemical pigments in the outer body layers, or the interaction between various mechanisms such as pigments amplifying iridescence in butterflies [1], produce a remarkable variety of colors with complex and diverse patterns in insects. These color patterns play important roles in the biology of insects, for example as secondary sexual characters [2], in aposematism and mimicry [3, 4], and crypsis [5].

Compared with Lepidoptera and Coleoptera, there are relatively few studies of color patterns in the order Hymenoptera. Studies are mostly related to the yellow and black patterns in Vespidae [6, 7] and the variation in color patterns in *Bombus* Latreille [8]. In Hymenoptera there are also few studies regarding the chemical and physical nature of the color patterns exhibited, some of the pigments reported being pheomelanins (black, orange-red, yellow variations in velvet ants and bumble bees) [3], eumelanin (brown-colored cuticle) and xanthopterin (yellow-colored cuticle) in *Vespa orientalis* Linnaeus [9]. Various scelionid wasps [10], principally neotropical species ranging from 2 to 10 mm in length, show a recurring color pattern of black head, orange mesosoma, and black metasoma (BOB pattern; Fig 1), which has also been documented in species of numerous other families of Hymenoptera [11]. Several lines of evidence suggest that this color pattern has evolved many times independently within scelionids: the phylogenetically most basal scelionids [12] do not show this pattern; though quite common in the Neotropics only a minority of species show this pattern; species showing the BOB pattern occur in genera that are widely scattered in the phylogeny and not all species in these genera have this pattern [11]. This in turn strongly suggests that the BOB color pattern has some biological function but there are currently no studies addressing this question. Nor

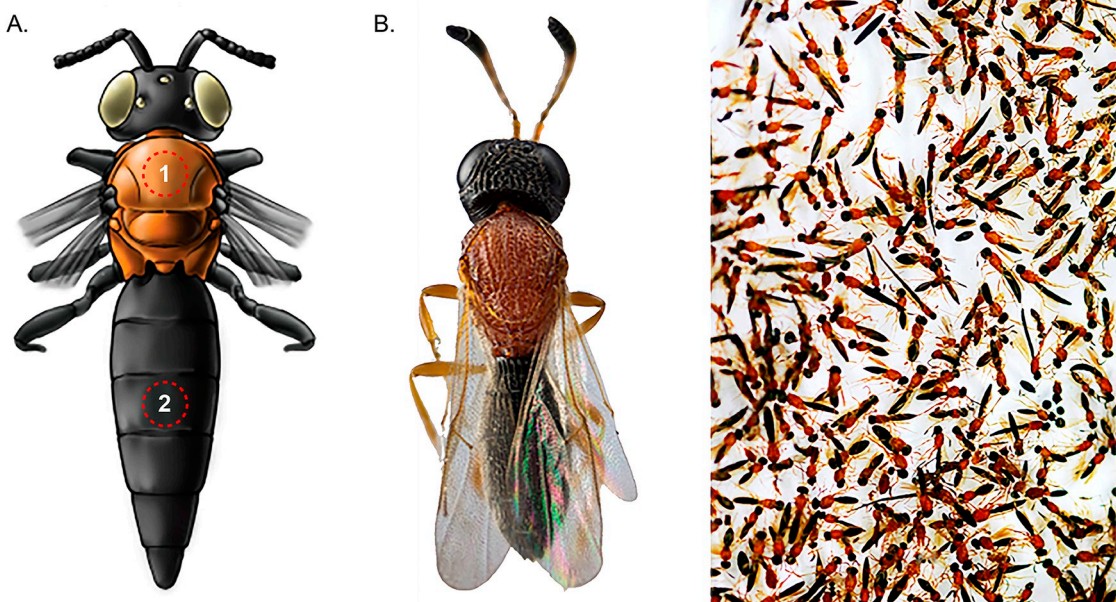

**Fig 1. BOB pattern illustration.** (A) Schematic detail of the area (dotted circles) used for measurements of both colors, zone of 1 mm$^2$ black of mesoscutum (1) and third tergite of metasoma (2). (B) Focus-stacked macro photography of a scelionid wasp with blak-orange-black color pattern obtained with a Reflex 850 camera coupled to a 20x microscope lens. The photograph at the far right, shows the striking black-orange-black color pattern of a group of scelionid specimens recently collected from the field; it was obtained with a Canon SX10 IS camera and a 105 mm macro lens.

have the orange and black colors present in these wasps been analyzed, which is the primary objective of the present investigation.

Color characterization of specimens, especially in taxonomy, has been done mostly subjectively, although more quantitative methods have been proposed [13]. There are few reports treating the measurement of color rigorously or analyzing spectral data and its variability as a functional measurement, which potentially can show differences detected by insects, and shed a light on their biology.

Among many studies that measure and compare colors in insects, the methods vary widely [7, 14, 15]. Endler et al. [16] took into account the continuous nature of the wavelengths as measures of color and used Principal Component Analysis (PCA) on the radiance spectra to analyze patterns of color, while recognizing its limitations, specifically, those related to the violation of assumptions for statistical tests to discriminate between colors. Functional Data Analysis (FDA) is a powerful method to study spectral data since it assumes that the underlying process generating the data is smooth, and hence provides tools to analyse a spectral measurement as a function and not as a series of data points [17]. For example, Rivas et al. [18] proposed the use of FDA techniques to test color changes in stone, and present a comparison with a multivariate method.

## Materials and methods

All experimental protocols were carried out in accordance with the involved institution's guidelines and regulations, thus, a scientific-academic license was obtained by the Ministry of Environment and Energy in relation to the National System of Conservation Areas, document N˚ ACTo-050-18, for the sampling and collection of the material under study.

A statistical approach based upon reflectance measurements of the dorsal surface of eight scelionid genera and 1 genus from a distantly related family (Evaniidae) was used in order to study the BOB color pattern. The reflectance spectra were measured by means of a microspectrophotometer. A statistical description of the BOB color pattern was obtained by means of different statistical approaches, applied to the spectral information as a function of genus, color, spots of measurement and specimens. This approach was used due to the small size of these wasps and the difficulty involved in obtaining sufficient fresh material for chemical analysis of the pigments.

### Microspectrophotometry

The reflectance spectra of 8 genera belonging to the hymenopteran family Scelionidae were measured with a 508 PV UV-Visible-NIR Microscope Spectrophotometer (CRAIC, Los Angeles, USA). The spectrophotometer is coupled to an Eclipse LV100 ND Microscope using episcopic illumination (Nikon, Tokyo, Japan). A spot size area of 200$\mu$m x 200$\mu$m was measured. A spectralon standard was used as a white reference because of the diffuse reflection produced by the roughness of the surface of the cuticle. Table 1 includes further information regarding the measurement settings.

### Specimens and sampling for optical measurements

Recently collected specimens are crucial to ensure the good condition of the external cuticle structures. Since Scelionidae are small in size and their hosts (insect eggs) are difficult to find, collecting specimens requires considerable effort.

Specimens of 8 genera were collected: *Acanthoscelio* Ashmead, *Baryconus* Foerster, *Chromoteleia* Ashmead, *Macroteleia* Westwood, *Opisthacantha* Ashmead (*Lapitha* Ashmead is currently considered to be a junior synonym [19]), *Scelio* Latreille, *Sceliomorpha* Ashmead,

**Table 1. Information about the analysis of color data.**

| Microspectrophotometry technical details | Information |
|---|---|
| Light source | 12 V-100 W halogen lamp |
| Magnification | 5X, CFI60 2 TU Plan Fluor BD, N.A 0.15, WD 18.0 mm, (Nikon,Tokyo, Japan) |
| White reference | Spectralon Diffuse Reflectance Standard USRS-00-010(Labsphere, North Sutton,NH, USA) |
| Dark reference | Internal attenuators |
| Software for spectral capture | Lambda Fire (CRAIC,Los Angeles,USA) |
| Integration time | 150 ms |
| Number of spectra averaged | 100 |

*Triteleia* Kieffer. They were collected with entomological nets over a period of 12 months. Additionally some specimens of the genus *Evaniella* Bradley (Evaniidae) were included, because they exhibit a similar BOB pattern, but are quite far removed from scelionids in the phylogeny of Hymenoptera [20].

The collection site was a patch within the protected zone of El Rodeo, which is located in the Colon district of the San José province, between the geographical coordinates 9˚ 52´–9˚ 56´ N and 84˚ 14´–84˚ 20´ W (Abra 3345 I and Río Grande 3345 IV cartographic sheets, National Geographic Institute 1989), at a distance of 18 km in a straight line from the capital city. The natural boundaries of the area are the Virilla River to the north, the Jaris river to the southeast and the Quebrada Honda river to the northeast. This natural reserve is composed of a secondary forest and a remnant of primary forest (approximately 200 ha). Specimens were identified by R.M. and P.H. using the keys to the scelionid genera [19] and voucher specimens were deposited in the Museum of Zoology of the University of Costa Rica. Due to the lack of taxonomic studies for six of the eight genera, species level identifications were not carried out. A previous study demonstrated that the vast majority hymenopterans with the BOB pattern, and those that are all black, do not show sexual dimorphism in color, at least in specimens that have been identified to species level in museum collections [11].

The specimens were killed by freezing and then, using a needle cut and polished on both ends, one square millimeter or a small notch of area was dissected in the mesoscutum (orange) and in the third tergite of the metasoma (black) (Fig 1). The orange and black samples of 4 specimens of each of the 8 genera were analyzed. For each specimen three samples were taken from the mesosoma (orange) and three from the metasoma (black).

## Colorimetric differences

Spectral reflectance curves can be studied in terms of colors using a geometrical representation or color space. The CIE color spaces are recognized to have important characteristics such as being device-independent and having a perceptual linearity [21]. Nevertheless, not all color spaces are suitable for comparing colors because distances between coordinates are not easily quantifiable due to the lack of uniformity of the space. For these effects, the CIE recommends using CIELAB color space due to its uniformity [22]. The corresponding coordinates are calculated using the procedure described in S1 Appendix.

The CIELAB color space is Euclidean, therefore distances between points can be used to represent approximately the perceived magnitude of color differences between object color

stimuli of the same size and shape, viewed under similar conditions. The CIE 1976 L, a, b (CIE-LAB) color difference definition was adopted [22]:

$$\Delta E = \sqrt{\Delta L^2 + \Delta a^2 + \Delta b^2} \tag{1}$$

Here, the quantities $\Delta L$, $\Delta a$, $\Delta b$, represent the subtraction of the corresponding coordinates of the two colors being compared and $\Delta E$ represents the distance between the two colors within the CIELAB space. There are other CIE definitions for color differences, see for example the ones explained in reference [22]. Nevertheless, as discussed by Melgosa [23], the definition in Eq 1 yields results that are not very different from more recent and complex definitions. Since the purpose of this work is not to evaluate the color space metrics, but to consolidate a quantitative framework to compare colors using the information available from microspectrophotometry, the CIE 1976 definition (Eq 1) was chosen for the sake of simplicity. Regarding the definition of a color difference threshold, it can be argued that CIE color spaces are based on human vision sensibility, and the color difference threshold will depend not only on the spectral region but also on the observer. Nevertheless, in order to establish a mean threshold value to define whether two colors are different, we used the value 2.3 as reported by [24] specifically for CIELAB. In this sense, when the color difference $\Delta E$ is less than this value, colors are considered as equal. Otherwise, colors are said to be different.

**RGB spectral components.** When calculating $\Delta E$, all the spectral information is condensed into a single number and it is not clear whether the difference comes from shorter or longer wavelengths. In this sense, the use of a spectral space instead of a color space would provide a more objective description of the physical variation among stimuli and also allows the identification of different physical mechanisms producing the same perception of color but having different spectral behavior [25]. Therefore, in order to visualize the characteristics of the the reflectance curves in different regions of the electromagnetic spectrum, we propose a trichromatic spectral space defined as follows using the CIE color matching functions (CMFs):

$$R(\lambda) = \frac{\bar{x}(\lambda)Z(\lambda)}{\int \bar{x}(\lambda)Z(\lambda)d\lambda}, \tag{2}$$

$$G(\lambda) = \frac{\bar{y}(\lambda)Z(\lambda)}{\int \bar{y}(\lambda)Z(\lambda)d\lambda} \tag{3}$$

and

$$B(\lambda) = \frac{\bar{z}(\lambda)Z(\lambda)}{\int \bar{z}(\lambda)Z(\lambda)d\lambda}, \tag{4}$$

where $Z(\lambda)$ is the measured reflectance, $\bar{x}(\lambda)$, $\bar{y}(\lambda)$ and $\bar{z}(\lambda)$ are the CIE 1976 CMF's [22].

We label these components as Red, Green and Blue (RGB) because their maximum amplitudes occur at 600 nm, 555 nm and 446 nm respectively.

It is recognized that spectral spaces have the advantage of providing information about the different processes generating phenotypic color diversity [25], and therefore we propose this RGB spectral component method as an alternative for the case when a direct chemical characterization can not be performed.

## Statistical analysis

**Data sets.** Each of the observations has 1185 recorded points: the reflectance ($Y$) as a function of wavelength value ($\lambda$), where $\lambda$ is a sequence of 1185 equidistant points from 420.2 nm to 919.7 nm. The use of ($\lambda$) instead of the usual ($s$) in FDA, is done to match the representation of a wavelength in the physics literature. For the remainder of this paper, the 1185 points will be called reflectance curves, which were recorded in the following way: each curve was constructed with the reflectance of color in mesoscutum and metasoma spots per colored area, and was measured three times in each specimen, in different spots. The observations correspond to the reflectance measured in the orange mesoscutum areas versus the black metasoma areas for 36 specimens: 4 specimens for each of the 8 genera, and 4 specimens for *Evaniella*, which was observed as a control (the latter genus belongs to a different family, which is not closely related to scelionids). The response variable is either a difference of reflectance or a measure of reflectance, and it can be measured in three different levels:

- The univariate mean measure: $m_{ijk}$ where $m$ is the mean difference between black and orange measures for all possible wavelengths in curve $ijk$, where $k$ represents repetitions, $j$ specimens, and $i$ genera. This is is used as a naive base comparison.

- The functional measure: $Y_{ijk}(\lambda)$ where the difference in reflectance between black and orange curves $Z(\lambda)$ is represented by $Y$, and was measured for each wavelength $\lambda$, and repeated $k$ times in each specimen $j$, and genera $i$. In this case, $Y_{ijk}(\lambda)$ describes a reflectance difference curve.

- The multivariate distance measure: a Euclidean distance $\Delta E$ as defined in 1 for each pair of curves being compared.

A table was created relating the different factors (spots, specimen, and genera) with the response in the univariate and functional case. In total, 96 combinations were recorded. In the multivariate distance case, a total of $N = C_2^{192} = 18336$ comparisons were used as observations, and then it was recorded whether or not each pair had a difference: genus, color, spot and specimen.

**Data analysis.** Prior to the analysis, descriptive statistics were generated for each genus and color section (black and orange). The objective of the data analysis is to contrast the null hypothesis of no difference between color reflectance means of each combination of genus and color area, and for that, univariate and functional ANOVA were performed. The multivariate distance case was also analyzed to match the physics literature on color differences. The following methods were used in each case:

- Univariate method: A one-way ANOVA was used to compare the mean difference between black and orange for all possible bandwidths $m_{ijk}$ with genus as the factor and controlling for specimen and spot.

- Functional method: A FANOVA [17] was used to compare the difference curves $Y_{ijk}(\lambda)$ with genus as the factor, and controlling for specimen and spot. The statistical F tests were included to interpret the contrast results for the mean difference curves. R packages `erpFtest` [26], and `fdANOVA` [27] were used.

- Multivariate distance method: A two-way ANOVA was used to compare the Euclidean distance $\Delta E$ between each combination of color and genus, using dichotomous variables as factors: color as 0 if the pair is from the same color, 1 if it differs; genus as 0 if the pair is from the same genus and 1 if it differs. Here also, specimen and spot were included as factors in the linear equation.

The objective is to illustrate the functional method compared to a univariate analysis approach, and with a multivariate distance alternative, to compare the results when taking into account all the information from the difference curves. More specifically, for the univariate and functional responses, the ANOVA model can be written as the following:

$$Y_{ijk} = \mu + \alpha_i + \beta_j + \epsilon_{ijk}, \tag{5}$$

for $i$ = 1, 2, ..., 8 genera, $j$ = 1, 2, ..., 4 specimens, and $k$ = 1, 2, 3 sampled spots in each colored area from each of the 4 specimens. Usual restrictions and assumptions for the ANOVA model apply:

$$\sum_{i=1}^{8} \alpha_i = 0, \qquad \sum_{j=1}^{4} \beta_j = 0, \qquad \epsilon_{ijk} \sim N(0, \sigma^2). \tag{6}$$

In the model, the global mean $\mu = 0$ given that we are working with difference curves, and $\alpha_i$ is the genera effect of level $i$, and $\beta_j$ is a block effect to explain the intra-genus variance. To account for the dependence between spots from the same specimens (in the two colored areas), a random effect $\eta_k$ was tested, and different covariance structures for $\Sigma_\eta$ were fitted (assuming dependence $\eta_k \sim N(0, \Sigma_\eta)$ or independence between spots). Lastly, $\epsilon_{ijk}$ is the residual that accounts for the unexplained variation specific to the $k$th observation, $i$th genus and $j$th specimen. Depending on the response $Y_{ijk}$, the residuals $\epsilon_{ijk}$ are a number (univariate) or a curve (functional). Assumptions were tested for all the model options.

For the multivariate distances, the ANOVA model differed in the factors:

$$\Delta E_i = \mu + \gamma_i + \eta_i + \omega_i + \rho_i + \gamma_i * \eta_i + \epsilon_i, \tag{7}$$

where the global mean $\mu$ represents the average change for the $N$ comparisons, and then each of the factors add the change given that the pair has different genera ($\gamma_i$), color ($\eta_i$), spots ($\omega_i$) or specimens ($\rho_i$) for each pair $i$. Finally, $\epsilon_i$ is the error modeled as white noise. Assumptions were tested for this model as well.

Statistical analysis was performed using the computing environment R [33]. The code and data to perform each of the tests mentioned in this paper are available for download in https://github.com/malfaro2/Mora_et_al. Data formatting and figures were prepared using the collection of R packages `Tidyverse` [28].

## Results

Reflectance curves in Fig 2 show differences both when comparing curves measured for the same genus and curves representing the reflectance of different genera. At the same time, such differences present variability within the group of measurements performed. For both between and within genus cases, these reflectance differences are statistically compared to zero in both the univariate data and the functional data (FDA method) (p-value < 0.00001 in all cases). The result is supported by the multivariate distance test, where there is evidence to say that overall change $\mu$ is greater than zero (p-value < 0.00001). Details about the significant factors that explain that change are summarized in the following sections.

### Overall differences

Fig 2 presents descriptive statistics for measured spectra for each genus and each of the spectral colors. There are some genera with more variability, and there is a tendency to have low reflectance at lower wavelengths for both black and orange spots. It is important to point out that the measurements were performed grouping genera and not species, a factor that may account

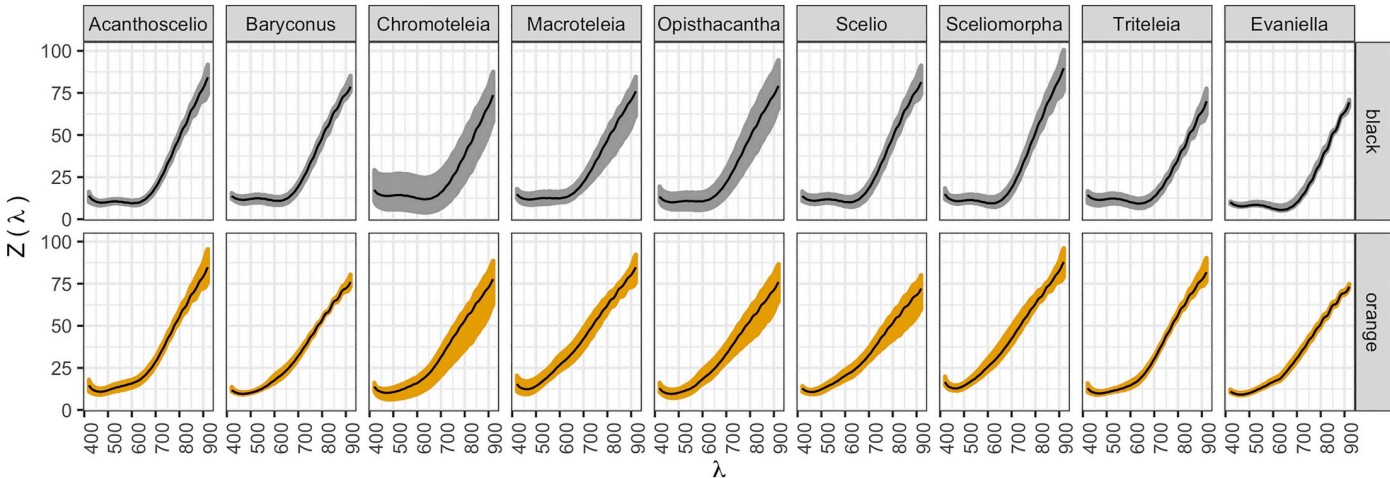

**Fig 2. Reflection functions.** Functions $Z(\lambda)$ in y axis, for each color within each genus. Black lines show the functional mean per case.

for the variability for some genera such as *Chromoteleia*. Because the spot size of 200 $\mu$m x 200 $\mu$m used in the reflectance measurements is more than 20 times larger than the features that comprise the mesosoma, the effect of topography of the cuticle and homogeneity of the pigment distribution (see S1 Fig), can partly explain the data dispersion in some of the genera.

The difference curves for reflectance spectra, defined as $Y_{ijk}(\lambda)$, are plotted in Fig 3. The patterns from Fig 2 appear more evident. For each genus, the behavior of the difference curves between black and orange can be described according to the wavelength in the following way. Curves above the black line mean that the black curves have higher reflectance values than the orange ones. Before 550 nm, differences are around zero, meaning that the reflectance values

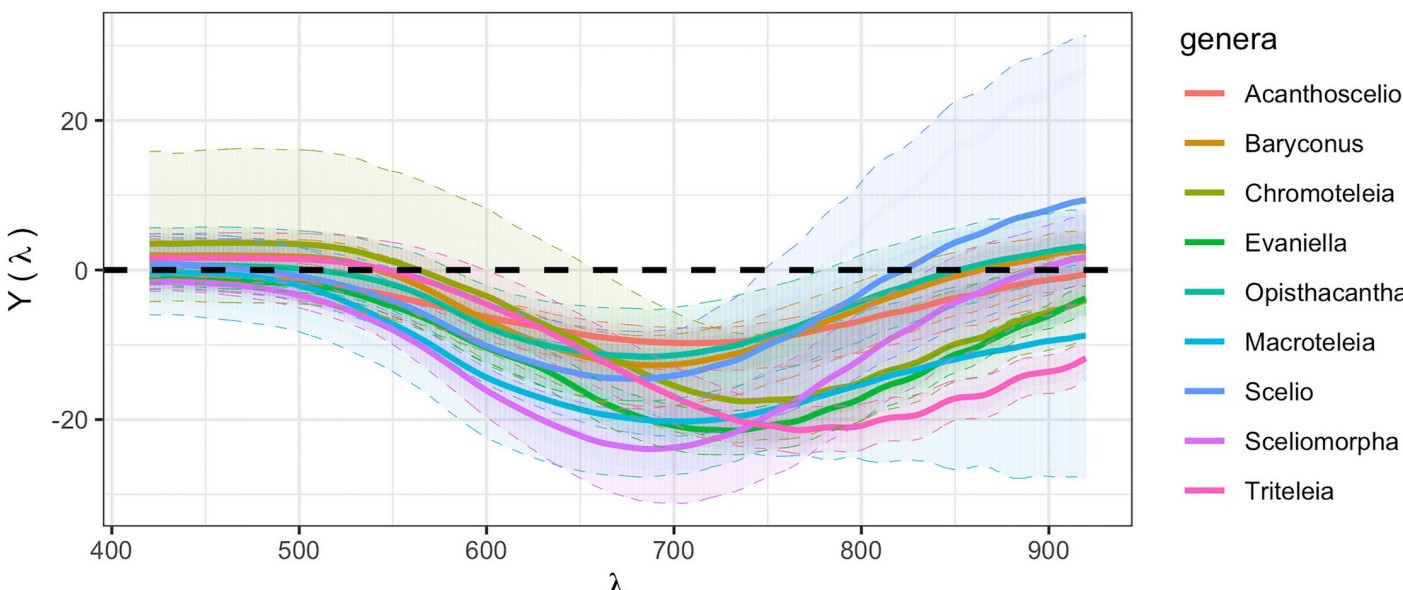

**Fig 3. Difference (black-orange) of mean reflectance functions in each genus.** Solid lines represent functional means, colored areas represent the area where difference functions in each genus are observed. A black dashed line represents the null hypothesis of no difference between colors in each genus. Curves above the black line mean that the black curves have higher reflectance values than the orange ones.

**Table 2. Mean reflectance $m_i$ for all possible bandwidths, by genus.** Genera are ordered according to the difference magnitude. [**] The univariate difference is not significantly different from zero. [a,b,c] Represent a group different from the rest, when doing univariate multiple comparisons. All analyses use $\alpha = 0.05$. Color representation corresponds to the CIE 1976 (L* a* b*) color space coordinates calculated from the mean reflectance curves from Fig 1.

| Genera | Black | Orange | Absolute Difference |
|---|---|---|---|
| *Baryconus* | 29.5307 | 33.692 | 4.1613[**,a] |
| *Opisthacantha* | 29.5492 | 33.7317 | 4.1825[**,a] |
| *Scelio* | 29.3678 | 33.8015 | 4.4337[**,a] |
| *Acanthoscelio* | 29.2331 | 34.3834 | 5.1503[a] |
| *Chromoteleia* | 26.5133 | 33.3825 | 6.8692[a,c] |
| *Triteleia* | 24.0089 | 34.1785 | 10.1696[b,c] |
| *Sceliomorpha* | 30.683 | 41.7377 | 11.0547[b] |
| *Macroteleia* | 29.5722 | 41.3887 | 11.8165[b] |
| *Evaniella* | 21.2445 | 32.3144 | 11.0699[b] |

are similar between orange and black in this region. Between 550 nm and 750 nm, the differences are below zero and are similar between genera. And finally, for wavelengths greater than 750 nm differences are both above and below zero and are dispersed with respect to genus. This analysis cannot be done if only the univariate measures are studied, given the information reduction as presented in S2 Fig.

In order to test the null hypothesis of $H_0: \alpha_i = 0$, whether or not the difference in reflectance between black and orange differs among genera, models were used to fit the data for each response. The best fit in all cases was from the models that assume independence between spots on the same specimen, so linear models were used subsequently. The univariate (p-value < 0.0001) and functional (p-value < 0.0001) results coincide in that in at least one genus examined, the mean reflectance of the orange spot is significantly different from the mean reflectance of the black spot, controlling for specimen and spot.

The statistical tests confirm what the descriptive plots showed. For each genus, the difference between black and orange is significantly different from zero for both the univariate and functional (FDA) case. Results from the functional ANOVA and a permutation test based on a Fourier basis function representation (`fdANOVA`), to test whether there is a mean difference between colors and genera, show that the calculated statistic is $T = 18.66961$ [26] with an associated p-value <0.0001, which means that the mean difference between the black and orange spots in at least one genus is different from the mean difference between the black and orange spots in the other genera, as shown in Fig 3. The univariate differences between genera shown in Table 2 only take into account the aggregated data collected at all wavelengths considered. The results for univariate comparison between genera can be found in S2 Appendix.

## Multivariate distance analysis

ANOVA results for the model of Eq 7 presented in Table 3, show how the overall average change in all comparisons is 5.0535 units with a standard error of 0.2642, which makes the average statistically greater than the 2.3 threshold value. The estimate of $\eta$ is 26.2081, which means that keeping all the other factors constant, the average $\Delta E$ (the distance between the two colors within the CIELAB space) is 26 units greater than the overall mean when the curves differ in color. Also, if the curves differ in both color and genus, the overall mean $\Delta E$ is 5.0535 + 4.3540 + 26.2081 + −4.3814 = 31.2342. The differences due to specimens are statistically not significant and the mean reflectance difference between measurements done in the same specimen but in different spots are on average 0.85, which makes it statistically significant but negligible on average, compared to the threshold.

**Table 3. ANOVA Results for model 7.**

| | Estimate | Std. Error | t value | p-value |
|---|---|---|---|---|
| $\mu$ | 5.0535 | 0.2642 | 19.12 | 0.0000 |
| $\gamma$ | 4.3540 | 0.2420 | 17.99 | 0.0000 |
| $\eta$ | 26.2081 | 0.3149 | 83.21 | 0.0000 |
| $\omega$ | 0.8524 | 0.1268 | 6.72 | 0.0000 |
| $\rho$ | -0.0425 | 0.1161 | -0.37 | 0.7142 |
| $\gamma:\eta$ | -4.3814 | 0.3358 | -13.05 | 0.0000 |

Given that it has been established that there is evidence of a difference between colors and genera in all analyses, there is a need to clarify which genera and which colors are making the difference. For that, a comparison matrix for $\Delta E$ was constructed and presented in two ways: comparisons between genera per color difference and comparisons between colors per genus difference. Such results are presented in Figs 4 and 5 respectively. As an example, Fig 4 depicts how the distance $\Delta E$ between *Triteleia* and *Acanthoscelio* curves presents two distributions: one when both have the same color (in vermilion) and another when they have different colors (in bluish green). Both distributions present relatively low variability, and the first distribution is closer to zero, although its box— 50% of its data— is not close to zero or the threshold. In contrast, the distance $\Delta E$ between *Acanthoscelio* curves for the same color and different specimens (in vermilion) clearly include the threshold and can be taken as not significantly different.

The difference between distributions is unclear when comparing colors. Fig 5 presents how in the case in which the colors differ (upper right corner) both distributions for $\Delta E$ are both clearly different from zero, while the distributions for $\Delta E$ of the curves with the same color are more difficult to distinguish from zero, especially if they are from the same genus (in vermilion).

The differences between curves can be described as an average of all $\Delta E$, for each genus and color. By that metric, both the maximum mean difference and the minimum mean difference can be established, as described in Table 4 for comparisons between curves of different colors and Table 5 for comparisons between curves of the same color, pointing to the top ten comparisons with wider differences. Similarly, top ten comparisons of different genera and the same genus are presented in S1 and S2 Tables.

## RGB spectral components

The comparison of the color component curves red ($R(\lambda)$), green ($G(\lambda)$) and blue ($B(\lambda)$), was done in terms of distance:

$$Distance = \sum_{\lambda} |R_c(\lambda) - Rc\prime(\lambda)|$$

for each pair of comparisons ($c$, $c\prime$). The results in Fig 6 show how the same differences depicted when using $\Delta E$ are very clear for the green and red components, but not for the blue component.

Panel (D) of Fig 6 shows the color component contributions for the *Acanthoscelio* and *Triteleia* genera for the black and orange measurements, respectively. The spectra were normalised by their own areas to illustrate the similarity of the contributions that are reflected in all data taken and presented in Figs 4 and 6 in panels A, B and C. Apart from the small changes in intensity for the normalised spectra, the maxima of each curve have a different dominant

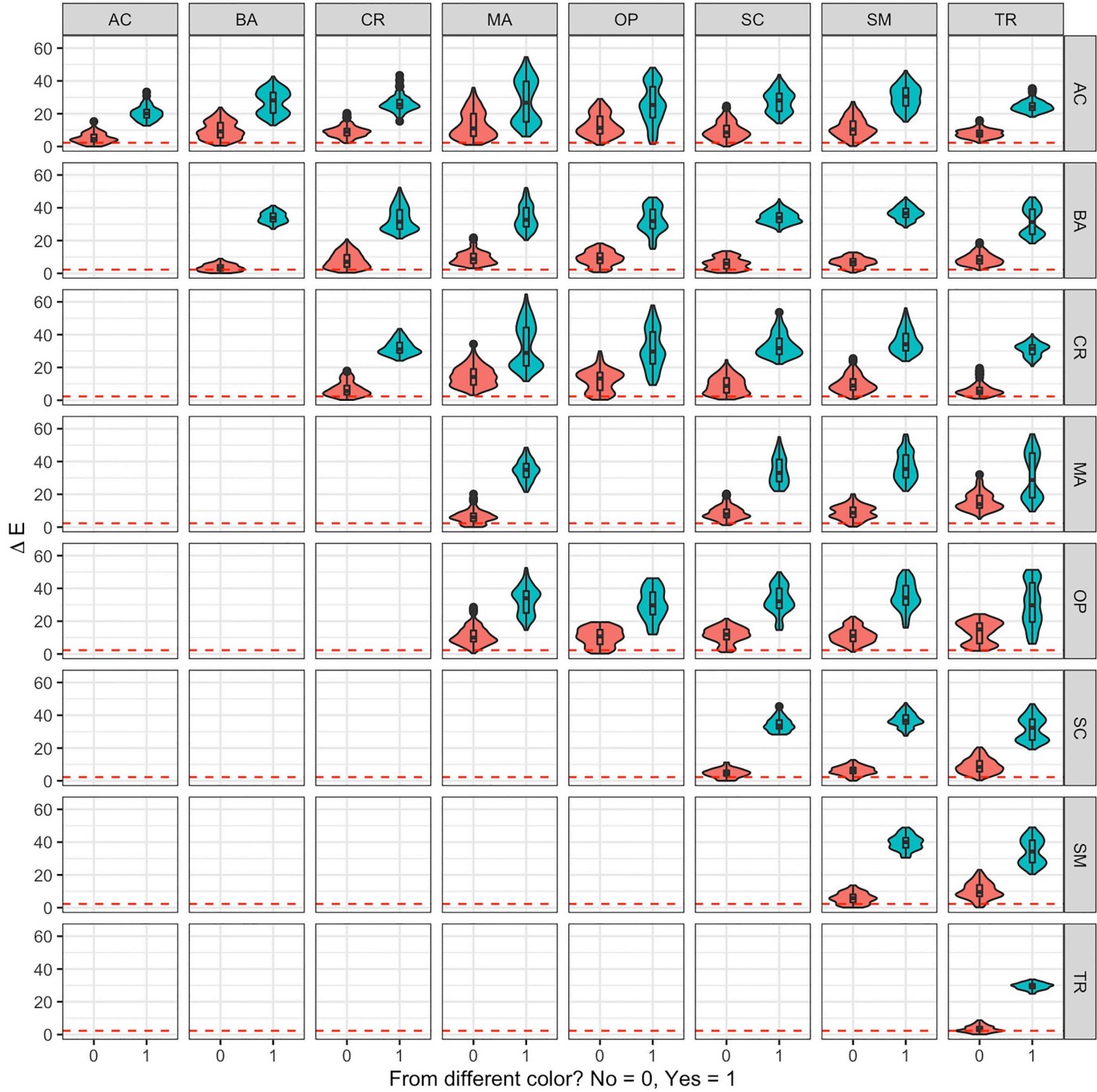

**Fig 4. Violin and box plot comparison per genus.** Distributions of ΔE calculated for pairs of curves of eight different genera: *Acanthoscelio* (AC), *Baryconus* (BA), *Chromoteleia* (CR), *Macroteleia* (MA), *Opisthacantha* (OP), *Scelio* (SC), *Sceliomorpha* (SM) and *Triteleia* (TR) and two scenarios: when they have the same color (in vermilion) and when their colors differ (bluish green). The dashed vermilion line represents the threshold of 2.3.

wavelength between genera, but the shape of the curves was very similar. The main changes are the differences in area when not normalised.

Mapping color variants in a spectral space can yield information about the relationship between genotypic space and evolution [29]. In this context, in order to visualize the data

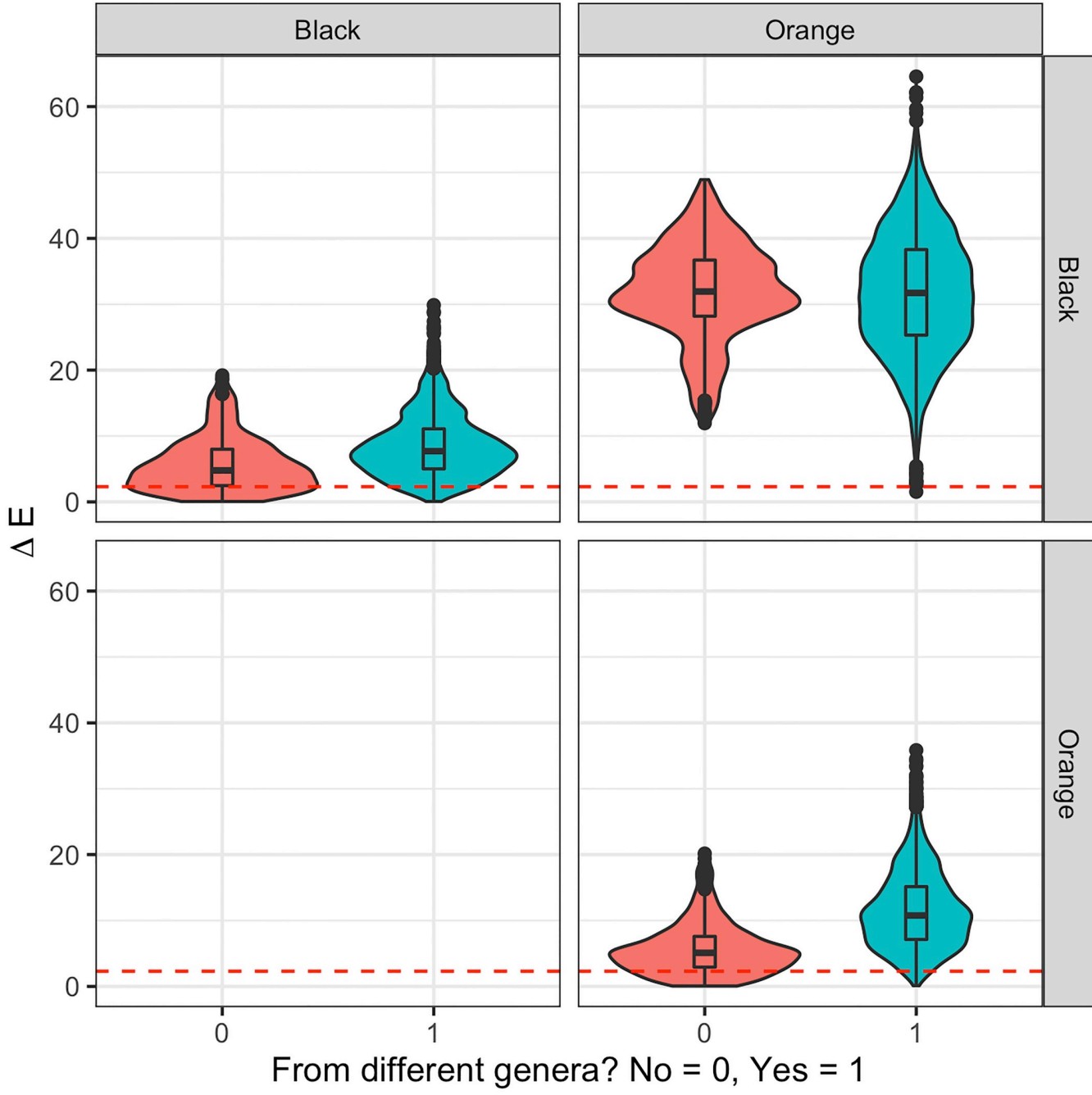

**Fig 5. Violin and box plot comparison per color.** Distributions of Δ$E$ calculated for pairs of two different colors, black and orange, and two scenarios: when they are the same genus (in vermilion) and when the genus differs (bluish green). The dashed vermilion line represents the threshold of 2.3.

provided by the spectral component analysis as a function of the genus, Fig 7 is presented. It shows the scatter plots for the black and orange measurements for all data, representing the logarithm of the area of each color component versus the dominant wavelength at the maxima of each contribution. The mean is plotted as a solid circle of the same color for each set of faded data points.

**Table 4. Top 10 according to $\overline{\Delta E}$ for comparisons of curves of different color.** The top 10 was extracted from a sample with overall minimum mean equal to 2.96 and maximum mean to 45.08. *Baryconus* (BA), *Chromoteleia* (CR), *Macroteleia* (MA), *Opisthacantha* (OP) and *Sceliomorpha* (SM).

| | Genus 1 | Genus 2 | Color 1 | Color 2 | $\overline{\Delta E}$ |
|---|---|---|---|---|---|
| 1 | CR | MA | BL | OR | 45.08 |
| 2 | MA | TR | OR | BL | 44.79 |
| 3 | MA | SM | OR | BL | 43.59 |
| 4 | OP | TR | OR | BL | 41.51 |
| 5 | CR | SM | BL | OR | 41.25 |
| 6 | SM | TR | OR | BL | 41.22 |
| 7 | MA | SC | OR | BL | 41.02 |
| 8 | CR | OP | BL | OR | 40.84 |
| 9 | SM | SM | BL | OR | 39.71 |
| 10 | BA | TR | OR | BL | 39.69 |

The range of wavelength values at the maxima for each color component contribution is interesting, being around 5 nm for the blue contribution in both black and orange measurements. For the green (black measurement) and red (orange measurement) components it is about 5 nm, but for the red (black measurement) and green (orange measurement) it is approximately 20 nm.

The genus *Evaniella* appears to separate from the group in the black measurements, but it is not that obvious for the orange measurements. Meanwhile, *Sceliomorpha* and *Macroteleia* group together for the green and red orange observations. The genus *Chromoteleia* is outside the main group in all three color components for the black observations and in the blue component of the orange observations. *Chromoteleia* and *Baryconus* also pair in the green and red components of the black and orange observations. Additionally *Sceliomorpha* and *Acanthoscelio* present a mean that is very close in the black measurements for the blue, green and red contributions. The genera *Opisthacantha* and *Triteleia* group together in the blue and green components for the orange contribution.

## Discussion

The following discussion is focused mainly on the contribution of the different statistical approaches to the physical and biological interpretation of the data obtained following the

**Table 5. Top 10 according to $\overline{\Delta E}$ for comparisons of curves of the same color.** The top 10 was extracted from a sample with overall minimum mean equal to 2.96 and maximum mean to 45.08. *Acanthoscelio* (AC), *Chromoteleia* (CR), *Macroteleia* (MA), *Opisthacantha* (OP) and *Sceliomorpha* (SM).

| | Genus 1 | Genus 2 | Color 1 | Color 2 | $\overline{\Delta E}$ |
|---|---|---|---|---|---|
| 1 | AC | MA | OR | OR | 19.64 |
| 2 | MA | TR | OR | OR | 18.26 |
| 3 | CR | MA | OR | OR | 17.02 |
| 4 | AC | OP | OR | OR | 16.95 |
| 5 | AC | SM | OR | OR | 15.77 |
| 6 | AC | BA | OR | OR | 14.35 |
| 7 | OP | TR | BL | BL | 14.16 |
| 8 | SM | TR | OR | OR | 13.69 |
| 9 | OP | MA | OR | OR | 13.61 |
| 10 | CR | OP | BL | BL | 13.32 |

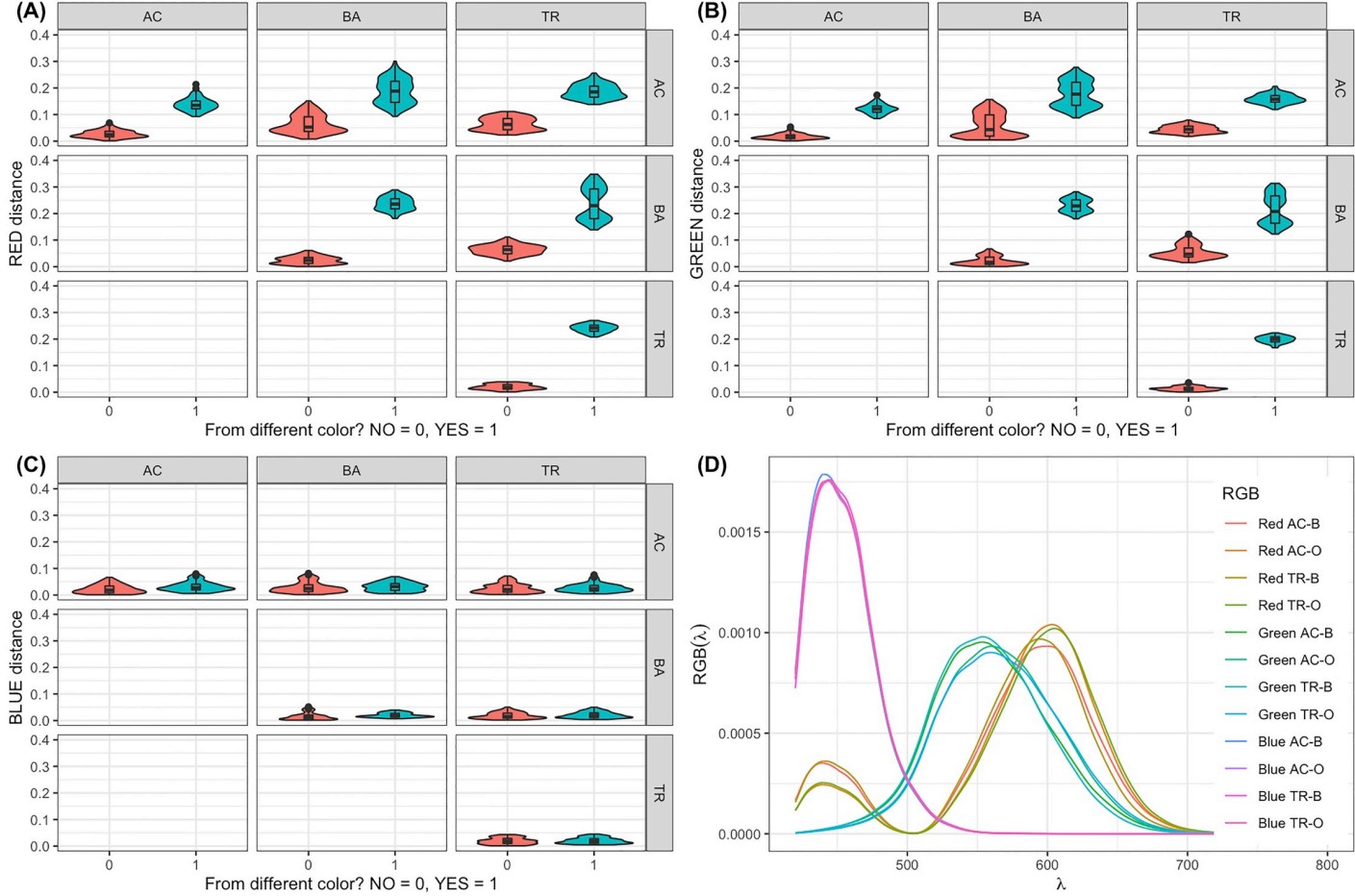

**Fig 6. Examples of violin and box plot comparison per genus and color component.** Distributions of *Distance* for (A) green, (B) red and (C) blue components, calculated for pairs of curves of three different genera: *Acanthoscelio* (AC), *Baryconus* (BA) and *Triteleia* (TR), and two scenarios: when they have the same color (in vermilion) and when their colors differ (bluish green). Plot (D) is an example of how the R, G, and B curves look for *Acanthoscelio* (AC) and *Triteleia* (TR) in orange and black spots.

procedures proposed, with the aim of establishing a quantitative analysis method for the study of cuticle color of insects. This section has two main parts: discussion about differences and suitability of the statistical methods and the interpretation of the physical results within a biological context.

## Suitability of the statistical methods

Regarding the suitability of the statistical methods applied, the present study highlights the differences that could arise using univariate methods, which ignore physical information contained in the reflectance measurement as a function of wavelength, versus analyzing data using FDA techniques.

The difference in intrageneric variances, confirms that the analysis of reflectance differences should not be done using univariate statistics. This variability is due to the difference in measurements across wavelengths: some sections are equal to zero, others are statistically different from zero, as described in the previous section. This can yield misleading results, finding univariate groups statistically similar, when the functional results are describing something different. In contrast, FDA provides methods for analyzing data that are believed to arise from

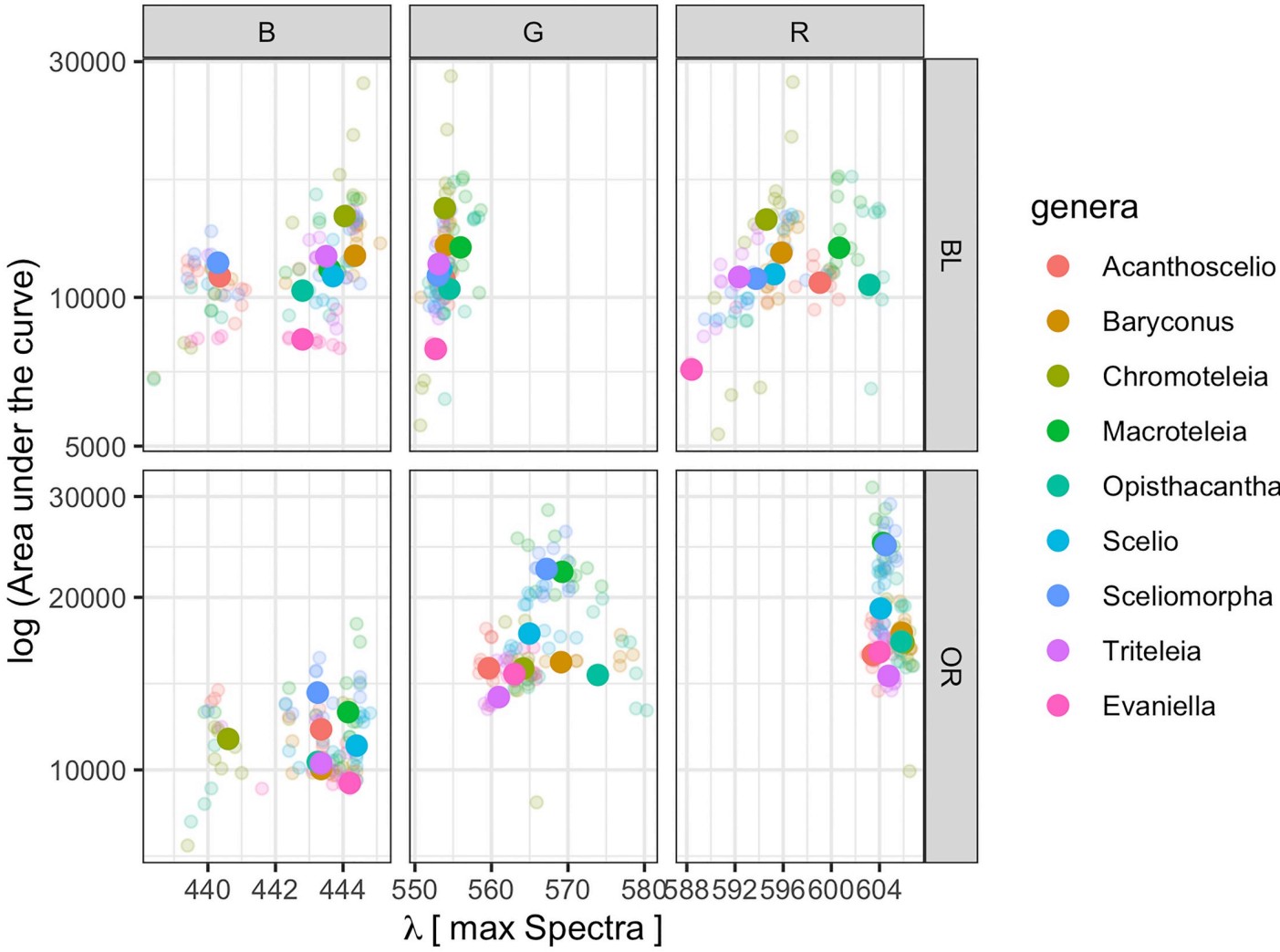

**Fig 7. Scatter plot for color components per genus.** Logarithm of the area under the curve, per component (red, green or blue) versus the corresponding wavelength in nm where the maximum occurs. Solid color points represent means for each genus and faded points correspond to independent observations. Plot (A) and (B) represent the components for the black and orange colors respectively.

curves evaluated at a finite grid of points allowing the incorporation of spectral information from the reflectance curves [17].

In addition, a multivariate Euclidean distance defined in terms of a quantity with physical meaning such as $\Delta E$, provides more evidence for analysis, besides confirming the FDA results. The details about multiple comparisons can be even more specific, if the color components are separated for each curve and the analysis is repeated.

The RGB component analysis showed that the nature of the differences found using the FDA and $\Delta E$ methods are related to the green and red color components, but no correlation was found for the blue component. This is consistent with results obtained qualitatively using the difference curves $Y_{ijk}(\lambda)$ shown in Fig 3. Moreover, ANOVA results for model 7 (Table 3) confirm the reproducibility of the experimental protocol and, for most of the cases studied, the results suggest that there is a low variability in the sample spectral measurements even though the taxonomic identification to just the genus level means that at least some of the genera could consist of more than one species.

The methodology described in this paper for statistically testing for color differences differs from those in the literature due to the level of scrutiny. It goes from the general question: "Is there a difference between colors?" to more specific contrasts between genera and color, using both the functional ANOVA and multivariate methods to compare color components. In this way, we are statistically comparing each pair of colors and genera, controlling for the other experimental factors.

### Physical results within a biological context

In general, the results of the spectral decomposition, together with the color differences as analyzed in Fig 5, suggest that the cuticle segments from different genera, but with the same color (black vs. black, orange vs. orange) might have a similar chemical composition or an identical chemical composition with different concentrations. Tonal variations could be related to different amounts of an individual pigment or a combination of similar polymers distributed in the cuticle.

Recent studies report that the color phenotype is associated with the concentration and ratio of pigment forms [3, 30], but other factors such as nanostrucutres of the integument can affect the expression of color [31]. In the case of pigments of different color (black vs. orange), the spectral blue components remain almost identical in terms of shape and characteristic wavelength (where the maximum occurs), suggesting that there is a common compound for the pigments. Notice that the normalization of the spectral components, Eqs (2), (3) and (4) allows the use of these features for comparison instead of intensity, which is related to the quantity of red, green or blue present in each color. The information given by the spectral components is important because optical properties are related to energetic processes due to electronic transitions when the pigment molecules interact with electromagnetic radiation. The shorter the wavelength, the more energetic the process is and therefore differences can be interpreted in terms of differences of electronic structure, which is related to the chemical composition of the pigments. A chemical characterization using infrared spectroscopy was attempted directly on the cuticle and results showed subtle differences suggesting a similar nature of the functional group attached to the molecule. Such results are not shown here because they are not conclusive due to the fact that the exact chemical composition of the cuticle for each color is unknown. A definitive identification of the pigment requires chemical extraction which at the moment could not be done because of the small amount of cuticle available.

*Evaniella* belongs to a completely different family (Evaniidae) and might therefore be expected to differ markedly from all the other genera. This indeed appears to be the case for its black color, but not its orange color (Fig 7). Nonetheless, within Scelionidae there are more differences in the orange color than in the black (Table 5). Comparing the results shown in Fig 7 with a recent phylogenetic reconstruction of scelionid genera [12], a few other relationships between spectral properties and phylogeny appear to be present. For example, *Baryconus* and *Chromoteleia* are paired in the green and red components for both black and orange, and these two genera are hypothesized to be closely related. *Opisthacantha* and *Triteleia* group together in the blue and green components of the orange coloration, and these two genera are also closely related. On the other hand, the orange color of *Macroteleia* and *Sceliomorpha* group together for the green and red components (Fig 7B) despite these genera not being closely related. These results are intriguing and merit further investigation to determine whether certain spectral properties parallel the phylogeny.

In some other insects, contrasting black and orange color patterns are known to serve as aposematic (warning) coloration for potential predators [32] and it is possible that the same is

true of the BOB pattern, although this has not yet been tested. However, the orange mesosoma becomes less common at higher elevations [11], suggesting that other factors such as thermo-regulation (black absorbs more sunlight) might play a role at these elevations; it is also possible that predators or aposematic models vary with elevation.

There is obviously much more to be learned about the black and orange colors present in numerous parasitoid wasps, but it is hoped that the results of the present spectral analyses provide a framework for future research, especially when a direct chemical characterization of the pigments is not possible and the optical and statistical methods proposed in this work can be applied.

## Conclusions

The present study provides the first analysis of the common black-orange-black pattern that is found in small (2-10 mm) parasitoid wasps. Because of their small size and difficulty in obtaining sufficient numbers of fresh specimens for pigment extraction, microspectrophotometry was used to obtain reflectance spectra for eight genera of scelionids. These reflectance data were analyzed by means of univariate analyses and by Functional Data Analysis, the latter proving to be superior because it gave a better representation of the physical information, given that it takes into account complete curves and not just summaries.

The RGB spectral components method is proposed for studying the origin of the color in the cases when a direct chemical composition is not possible. In the case of the genera studied, the spectral blue components of the orange and black color were found to be almost identical, suggesting that there is a common compound for the pigments. This result holds even when different genera were considered for comparison. Also spectral measures can and should be used for quantifying differences in biological color patterns since such analysis provides information that is not tinted by the illuminant.

A correlation between the mean values of characteristics of the color components was used in an attempt to group genera that show similar values and some, but not all, of these groupings are similar to their positions in a previously published phylogeny of the genera.

Further study, including a chemical analysis of the pigments and classification of specimens into species should be performed in order to obtain validation of these results.

## Supporting information

**S1 Appendix. Color coordinates calculation.**
(PDF)

**S2 Appendix. Univariate comparison for the inter-genera case.**
(PDF)

**S3 Appendix. Eosin hematoxylin technique.**
(PDF)

**S1 Fig. Eosin hematoxylin technique.** Cross section of the cuticle shows the pigment concentrated in the epicuticle, of a specimen of *Baryconus* with black-orange-black color (a) and black color (b). The variation in the distribution of the pigment together with the roughness of the cuticle surface can contribute to the reflection of light in different directions and therefore to the dispersion of the data.
(PDF)

**S2 Fig. Box plots for univariate mean differences per genus.** Black dashed line represents the null hypothesis of no difference between colors per genus. *Acanthoscelio* (AC), *Baryconus*

(BA), *Chromoteleia* (CR), *Macroteleia* (MA), *Opisthacantha* (OP), *Scelio* (SC), *Sceliomorpha* (SM), *Triteleia* (TR) and *Evaniella* (EV).
(PDF)

**S1 Table. Top 10 $\overline{\Delta E}$ differences for comparisons of curves of different genera.** The top 10 was extracted from a sample with overall minimum mean equal to 2.96 and maximum mean to 45.08 *Acanthoscelio* (AC), *Baryconus* (BA), *Chromoteleia* (CR), *Macroteleia* (MA), *Opisthacantha* (OP) and *Sceliomorpha* (SM).
(PDF)

**S2 Table. Top 10 $\overline{\Delta E}$ differences for comparisons of curves of the same genera.** The top 10 was extracted from a sample with overall minimum mean equal to 2.96 and maximum mean to 45.08. *Acanthoscelio* (AC), *Baryconus* (BA), *Chromoteleia* (CR), *Macroteleia* (MA), *Opisthacantha* (OP), *Scelio* (SC), *Sceliomorpha* (SM) and *Triteleia* (TR).
(PDF)

## Acknowledgments

This work was inspired by personal conversations with Lubomir Masner. We would like to give special thanks to Mauricio Arce for the macro photography and to Juan Porras Peñaranda for his support in eosin-hematoxilin analysis. We thank three anonymous reviewers whose comments and suggestions helped improve and clarify this manuscript.

## Author Contributions

**Conceptualization:** Rebeca Mora-Castro, Marcela Hernández-Jiménez, Paul Hanson-Snortum.

**Data curation:** Marcela Alfaro-Córdoba.

**Formal analysis:** Marcela Hernández-Jiménez, Marcela Alfaro-Córdoba.

**Funding acquisition:** Rebeca Mora-Castro, Esteban Avendano.

**Investigation:** Rebeca Mora-Castro.

**Methodology:** Rebeca Mora-Castro, Marcela Hernández-Jiménez, Marcela Alfaro-Córdoba, Esteban Avendano.

**Project administration:** Rebeca Mora-Castro, Marcela Hernández-Jiménez.

**Resources:** Rebeca Mora-Castro, Marcela Hernández-Jiménez, Marcela Alfaro-Córdoba, Esteban Avendano, Paul Hanson-Snortum.

**Software:** Marcela Alfaro-Córdoba.

**Supervision:** Marcela Hernández-Jiménez, Esteban Avendano, Paul Hanson-Snortum.

**Validation:** Esteban Avendano, Paul Hanson-Snortum.

**Visualization:** Marcela Alfaro-Córdoba.

**Writing – original draft:** Rebeca Mora-Castro, Marcela Hernández-Jiménez, Marcela Alfaro-Córdoba.

**Writing – review & editing:** Rebeca Mora-Castro, Marcela Hernández-Jiménez, Marcela Alfaro-Córdoba, Esteban Avendano, Paul Hanson-Snortum.

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
