## [Decision Letter · Decision Letter 0]

21 Jul 2019

PONE-D-19-14642

Spectral measure of color variation of black - orange - black (BOB) pattern in small parasitic wasps (Hymenoptera: Scelionidae), a statistical approach

PLOS ONE

Dear Mrs. Mora Castro,

Thank you for submitting your manuscript to PLOS ONE. After careful consideration, we feel that it has merit but does not fully meet PLOS ONE’s publication criteria as it currently stands. Therefore, we invite you to submit a revised version of the manuscript that addresses the points raised during the review process.

The submission represents a new window into coloration patterns in Hymenoptera (a lineage that is understudied in this regard relative to beetles or Lepidoptera), specifically by investigating several scelionid genera. While all three reviewers indicated that this manuscript is sound from a methodological standpoint, there is consensus that the text must be reworked. In particular, the authors must work to ensure that (i) the core focus and results of the paper are clear to the reader, (ii) that text reflects what is currently known about the biology and systematics of these wasps, and (iii) that the overall clarity of the paper be improved. See attached reviews for additional details in these areas, as well as line-by-line suggested improvements (including those embedded in the attached Adobe Acrobat PDF). If the authors choose not to incorporate some suggestions, these should be noted and justified in the "Response to reviewers."

We would appreciate receiving your revised manuscript by Sep 04 2019 11:59PM. To enhance the reproducibility of your results, we recommend that if applicable you deposit your laboratory protocols in protocols.io, where a protocol can be assigned its own identifier (DOI) such that it can be cited independently in the future. For instructions see: http://journals.plos.org/plosone/s/submission-guidelines#loc-laboratory-protocols

We look forward to receiving your revised manuscript.

Kind regards,

Phillip Barden

Academic Editor

PLOS ONE

Journal Requirements:

1. In your Methods section, please provide additional location information, including geographic coordinates for the data set if available.

2. To comply with PLOS ONE submissions requirements for field studies, please provide the following information in the Methods section of the manuscript and in the “Ethics Statement” field of the submission form (via “Edit Submission”):

a) Provide the name of the authority who issued the permission for each location (for example, the authority responsible for a national park or other protected area of land or sea, the relevant regulatory body concerned with protection of wildlife, etc.). If the study was carried out on private land, please confirm that the owner of the land gave permission to conduct the study on this site.

b) For any locations/activities for which specific permission was not required, please

- i. state clearly that no specific permissions were required for these locations/activities, and provide details on why this is the case

- ii. confirm that the field studies did not involve endangered or protected species

c) For vertebrate studies only, please provide the following additional information:

- i. Full details of collection and sampling methods, including method of sacrifice if applicable

- ii. State whether the vertebrate work was approved by an Institutional Animal Care and Use Committee (IACUC) or equivalent animal ethics committee. If no approval was obtained, please explain why it was not required.

- iii. State clearly whether all sampling procedures and/or experimental manipulations were reviewed or specifically approved as part of obtaining the field permit.

For more information about PLOS ONE submissions requirements for field studies, please refer to http://journals.plos.org/plosone/s/submission-guidelines#loc-animal-research.

3. We note you have included a table to which you do not refer in the text of your manuscript. Please ensure that you refer to Tables 6 & 7 in your text; if accepted, production will need this reference to link the reader to the Table.

Reviewers' comments:

Reviewer's Responses to Questions

**Comments to the Author**

1. Is the manuscript technically sound, and do the data support the conclusions?

Reviewer #1: Partly

Reviewer #2: Yes

Reviewer #3: Yes

2. Has the statistical analysis been performed appropriately and rigorously? 

Reviewer #1: I Don't Know

Reviewer #2: I Don't Know

Reviewer #3: Yes

3. Have the authors made all data underlying the findings in their manuscript fully available?

Reviewer #1: Yes

Reviewer #2: No

Reviewer #3: Yes

4. Is the manuscript presented in an intelligible fashion and written in standard English?

Reviewer #1: Yes

Reviewer #2: Yes

Reviewer #3: Yes

5. Review Comments to the Author

Reviewer #1: See attachment......................................................................................

Reviewer #2: I think that this paper begins to address a very interesting phenomenon in Hymenoptera. I believe that eventually it should be accepted, but not without revision to the manuscript. I am not a statistician, so I cannot give an informed analysis of the statistical methods. My area of expertise is in the systematics of Platygastroidea (which includes Scelionidae) and most of my comments are made regarding the biology and taxonomy of the wasps. There are many small grammatical errors that I have indicated on the attached pdf, as well as more substantive comments about incorrect or unsupported assertions. I selected "Major Revision", because I think that the authors have to be careful about some of the broad assertions in the paper. That said, they likely can be corrected quickly.

Reviewer #3: I recommend that this paper be published after minor amendments as outlined in the attached document. A couple of sections could be improved by being shortened with more concise writing and there are a few instances where the English needed improving.

6. PLOS authors have the option to publish the peer review history of their article (what does this mean?). If published, this will include your full peer review and any attached files.

Reviewer #1: No

Reviewer #2: No

Reviewer #3: No

---

## [Author Response · Author response to Decision Letter 0]

28 Aug 2019

Please see "Response to reviewers" document. We wrote all responses to the reviewers in the document 'Response to Reviewers', including the 6 questions labeled as "Comments to the Author". For each comment from the editor and reviewer, we have a response that includes specific changes that have been made. Also, all the figures and code have been corrected according to their suggestions, and new figure versions have been uploaded.

---

## [Decision Letter · Decision Letter 1]

18 Sep 2019

PONE-D-19-14642R1

Spectral measure of color variation of black - orange - black (BOB) pattern in small parasitic wasps (Hymenoptera: Scelionidae), a statistical approach

PLOS ONE

Dear Mora-Castro,

Thank you for submitting your manuscript to PLOS ONE. After careful consideration, we feel that it has merit but does not fully meet PLOS ONE’s publication criteria as it currently stands. Therefore, we invite you to submit a revised version of the manuscript that addresses the points raised during the review process.

Both reviewers have indicated that substantial and critical improvements have been made. However, the consensus is that the manuscript would benefit from an additional round of minor revisions. Please consider the specific comments made by both reviewers carefully and incorporate this feedback into your next submission. If the authors disagree with any particular suggestions or comments, these disagreements should be noted and justified in a response letter.

We would appreciate receiving your revised manuscript by Nov 02 2019 11:59PM. To enhance the reproducibility of your results, we recommend that if applicable you deposit your laboratory protocols in protocols.io, where a protocol can be assigned its own identifier (DOI) such that it can be cited independently in the future. For instructions see: http://journals.plos.org/plosone/s/submission-guidelines#loc-laboratory-protocols

We look forward to receiving your revised manuscript.

Kind regards,

Phillip Barden

Academic Editor

PLOS ONE

Reviewers' comments:

Reviewer's Responses to Questions

**Comments to the Author**

1. If the authors have adequately addressed your comments raised in a previous round of review and you feel that this manuscript is now acceptable for publication, you may indicate that here to bypass the “Comments to the Author” section, enter your conflict of interest statement in the “Confidential to Editor” section, and submit your "Accept" recommendation.

Reviewer #1: All comments have been addressed

Reviewer #2: All comments have been addressed

2. Is the manuscript technically sound, and do the data support the conclusions?

Reviewer #1: Yes

Reviewer #2: Yes

3. Has the statistical analysis been performed appropriately and rigorously? 

Reviewer #1: I Don't Know

Reviewer #2: I Don't Know

4. Have the authors made all data underlying the findings in their manuscript fully available?

Reviewer #1: Yes

Reviewer #2: Yes

5. Is the manuscript presented in an intelligible fashion and written in standard English?

Reviewer #1: Yes

Reviewer #2: Yes

6. Review Comments to the Author

Reviewer #1: The authors have addressed all of the major issues with their manuscript. In its current form, its utility to ecologists, evolutionary biologists, etc. will be limited by the technical and unclear language that the authors use. However that is not a fatal flaw.

Scelionidae are parasitoids, not parasites.

l 2, 455, etc. “(un)characterized” how? Some people might think that calling a color “orange” is a form of characterization.

l 5 says "2 to 10 mm" but l 40 & 456 say "3 to 10 mm", which is it?

l 12. "both inter and intragenera" -> "both between and within genera"

l 35, 39, etc. Need taxonomic authorities for genera and species.

Fig. 1 caption. Make the 2 superscript in "mm2".

l 74-5. What about the evaniid?

l 83. "Hymenopteran" -> "hymenopteran"

l 94. Need a space between "Acanthoscelio" and "Ashmead".

l 95. Need a space between "Westwood," and "Opisthacantha".

l 110, 111, 230. Remove the spaces before the periods.

l 115. Remove "by".

It's odd that Fig. 2 is cited in the Methods, since this figure shows Results of this study.

l 153. Italicize the E in Delta-E.

l 160. Remove "in this work, ".

l 167. Remove the second period.

l 178. "referred as" -> "called"

l 179. What about the metasoma?

l 214. Many journals prefer a monospace font (\\texttt{} in LaTeX) for the names of software packages, such as erpFtest and fdANOVA.

l 225. Shouldn't "species" be "genera"?

l 229. Should there be a comma between "i" and "and"?

l 230. Remove "In order".

l 233. "accounting" -> "that accounts"

l 242. Remove the parentheses ().

l 245. See my comment about l 214.

l 247. "provide support for" -> "support"

l 247. State the meaning of "reflectance difference variations between and within genera".

l 258. "in some cases" -> "for some genera"

l 260. "that" -> "than"

l 263. Put a comma after the closed paren.

l 267-8. Explain to the reader what it means when "differences are below zero".

l 269. Add a comma after "750nm".

l 272. Remind the reader of what this hypothesis means.

l 280. "are" -> "is"

l 282. Why isn't the "Fourier basis function representation" mentioned in the Methods?

l 285-6. Do the authors mean to say "from the MEAN difference between the black and orange spots in the other genera"?

l 311. Replace the hyphens with em dashes (--- in LaTeX).

l 312. Remove "and Acanthoscelio".

l 321. “maximum and minimum mean” is somewhat tortured/tortuous language. I’d rephrase this.

l 368. I really like this opening to the Discussion. I’d suggest using subheadings for the two subsections.

l 370. Remove “in terms of results”.

l 379. “Functional Data Analysis (FDA)” -> “FDA” you already defined this acronym.

l 395. “This particular method” make it clear which method you’re talking about, you mentioned various methods in the previous paragraph.

l 430-1. What do the authors mean by “the pigments are actually convergent” - did wasps in both families converge on use of the same pigment, or do they use two different pigments that produce colors with convergent reflectance properties?

l 563. Capitalize "B" in "batesian"

l 569, 592. Need a space before the open paren.

Reviewer #2: There are some grammatical corrections in the attached pdf. I think that the speculation about convergent evolution is unfounded and should be removed entirely.

7. PLOS authors have the option to publish the peer review history of their article (what does this mean?). If published, this will include your full peer review and any attached files.

Reviewer #1: No

Reviewer #2: No

---

## [Author Response · Author response to Decision Letter 1]

26 Sep 2019

Reviewer #1 highlighted in red color in manuscript.

Scelionidae are parasitoids, not parasites. DONE

l 2, 455, etc. “(un)characterized” how? Some people might think that calling a color “orange” is a form of characterization. DONE (changed to “have not been analyzed”, “analyzing”, etc.)

l 5 says "2 to 10 mm" but l 40 & 456 say "3 to 10 mm", which is it? DONE

l 12. "both inter and intragenera" -> "both between and within genera". DONE

l 35, 39, etc. Need taxonomic authorities for genera and species. DONE

Fig. 1 caption. Make the 2 superscripts in "mm2". DONE

l 74-5. What about the evaniid? DONE

l 83. "Hymenopteran" -> "hymenopteran" DONE

l 94. Need a space between "Acanthoscelio" and "Ashmead". DONE

l 95. Need a space between "Westwood," and "Opisthacantha". DONE

l 110, 111, 230. Remove the spaces before the periods. DONE

l 115. Remove "by". DONE

L 122 It's odd that Fig. 2 is cited in the Methods, since this figure shows the Results of this study. This line was replaced by “Spectral reflectance curves can be studied in terms of colors using a geometrical representation or color space.” Figure 2 reference was moved to the line after the first paragraph of Results, where is mentioned for the first time. DONE 

l 153. Italicize the E in Delta-E. DONE

l 160. Remove "in this work, ". DONE

l 167. Remove the second period. DONE

l 178. "referred as" -> "called" DONE

l 179. What about the metasoma? DONE

l 214. Many journals prefer a monospace font (\\texttt{} in LaTeX) for the names of software packages, such as erpFtest and fdANOVA. DONE

l 225. Shouldn't "species" be "genera"? DONE

l 229. Should there be a comma between "i" and "and"? DONE

l 230. Remove "In order". DONE

l 233. "accounting" -> "that accounts" DONE

l 242. Remove the parentheses (). DONE

l 245. See my comment about l 214. DONE

l 247. "provide support for". This paragraph changed, see next comment. DONE

l 247. State the meaning of "reflectance difference variations between and within genera". This statement was replaced by “Reflectance curves in Fig~\\ref{fig:F2} show differences both when comparing curves measured for the same genera and curves representing the reflectance of different genera. At the same time, such differences present variability within the group of measurements performed. For both between and within genera cases, this reflectance differences are statistically compared to zero in both the univariate data and the functional data (FDA method) (p-value $< 0.00001$ in all cases).” DONE

l 258. "in some cases" -> "for some genera" DONE

l 260. "that" -> "than" DONE

l 263. Put a comma after the closed paren. DONE

l 267-8. Explain to the reader what it means when "differences are below zero". DONE

l 269. Add a comma after "750nm". DONE

l 272. Remind the reader of what this hypothesis means. DONE

l 280. "are" -> "is" DONE

l 282. Why isn't the "Fourier basis function representation" mentioned in the Methods? It is the method performed by \\texttt{fdANOVA}, we added the clarification. DONE

l 285-6. Do the authors mean to say "from the MEAN difference between the black and orange spots in the other genera"? Correct, thank you for the correction. DONE

l 311. Replace the hyphens with em dashes (--- in LaTeX). DONE

l 312. Remove "and Acanthoscelio". DONE

l 321. “maximum and minimum mean” is somewhat tortured/tortuous language. I’d rephrase this. DONE

l 368. I really like this opening to the Discussion. I’d suggest using subheadings for the two subsections. DONE

l 370. Remove “in terms of results”. DONE

l 379. “Functional Data Analysis (FDA)” -> “FDA” you already defined this acronym. DONE

l 395. “This particular method” make it clear which method you’re talking about, you mentioned various methods in the previous paragraph.DONE

l 430-1. What do the authors mean by “the pigments are actually convergent” - did wasps in both families converge on use of the same pigment, or do they use two different pigments that produce colors with convergent reflectance properties? DONE. This sentence was eliminated as suggested by reviewer 2.

l 563. Capitalize "B" in "batesian" DONE

l 569, 592. Need a space before the open paren. DONE

Reviewer #2:

I think that the speculation about convergent evolution is unfounded and should be removed entirely. DONE

There are some grammatical corrections in the attached pdf. DONE

Thank you very much for the comment we did a detailed and careful reading and found 8 grammatical errors that were corrected.

---

## [Editor Report · Decision Letter 2]

1 Oct 2019

Spectral measure of color variation of black - orange - black (BOB) pattern in small parasitoid wasps (Hymenoptera: Scelionidae), a statistical approach

PONE-D-19-14642R2

Dear Dr. Mora-Castro,

We are pleased to inform you that your manuscript has been judged scientifically suitable for publication and will be formally accepted for publication once it complies with all outstanding technical requirements.

With kind regards,

Phillip Barden

Academic Editor

PLOS ONE

---

## [Editor Report · Acceptance letter]

8 Oct 2019

PONE-D-19-14642R2 

Spectral measure of color variation of black - orange - black (BOB) pattern in small parasitoid wasps (Hymenoptera: Scelionidae), a statistical approach 

Dear Dr. Mora-Castro:

I am pleased to inform you that your manuscript has been deemed suitable for publication in PLOS ONE. Congratulations! Your manuscript is now with our production department. 

With kind regards,

on behalf of

Dr. Phillip Barden 

Academic Editor

PLOS ONE